# Repair of Traumatic Urethral Strictures: La Paz University Hospital Experience

**DOI:** 10.3390/jcm12010054

**Published:** 2022-12-21

**Authors:** Cristina Ballesteros Ruiz, Carlos Toribio-Vázquez, Esaú Fernández-Pascual, Emilio Ríos, Andrea Rodríguez Serrano, J. M. Alonso Dorrego, Manuel Girón de Francisco, J. A. Moreno, Paloma Cárcamo Valor, Luis Martínez-Piñeiro

**Affiliations:** 1Urology Department, Hospital Universitario La Paz, Paseo de la Castellana 261, 28046 Madrid, Spain; 2Instituto de Investigación Hospital Universitario La Paz (IDiPaz), Calle de Pedro Rico, 6, 28029 Madrid, Spain

**Keywords:** urethral strictures, pelvic fracture, blunt perineal trauma, urethroplasty

## Abstract

Introduction: The management of traumatic urethral strictures remains a challenge for urologists. Alteration of the pelvic anatomy and the significant fibrosis generated by the trauma make surgical repair complex. In most cases, the existing defect between the urethral ends is small, and the ideal treatment is end-to-end perineal urethroplasty. Cases of extensive strictures that are left with long gap defects may require the use of different sequential maneuvers to achieve a tension-free anastomosis. Objective: To describe the experience at our center with urethral strictures induced by closed perineal trauma. Materials and methods: A retrospective analysis of 116 patients who underwent urethroplasty for urethral stricture after blunt perineal trauma at our center between 1965 and 2020 was conducted. Demographic data, date, mechanism of action of the trauma, emergency management, previous urethral interventions, surgical technique carried out in our center, complications, presence of erectile dysfunction, and urinary incontinence were collected. Results: 82 patients (70.7%) presented with pelvic fractures. The most frequent etiology of trauma was traffic accidents (68%), followed by crushing injuries (24%). Suprapubic cystostomy was placed in 50.2% of patients, and urethral realignment was performed in 25.3%. The mean stricture length was 2.2 cm, affecting mostly the membranous urethra (67%). During surgery, it was necessary to perform crural separation in 61.5% and partial pubectomy in 18.8% of the cases. Erectile dysfunction developed after trauma in 40.5% of cases, while new erectile dysfunction was noted in 4.3% of patients after surgery. Surgery was successful in 91.3% of cases, with a median follow-up of 16 (6–47) months. Conclusion: Delayed anastomotic urethroplasty offers a high success rate in traumatic urethral strictures.

## 1. Introduction

The treatment of traumatic urethral strictures remains a challenge for urologists. Alteration of the pelvic anatomy and the significant fibrosis generated by the trauma make surgical repair complex. Pelvic fractures are associated with male posterior urethra injury in 3.5–19% of cases. Female urethral involvement is far less frequent, presenting in 0–6% of cases and usually caused by direct contusion, laceration, or fractured bone fragments [1]. Urethral trauma usually occurs in young males and is caused by blunt trauma in 90% of cases [2].

The membranous and the proximal bulbar urethra are most frequently affected. The prostate is protected by the anterior pelvic ring and is attached to the symphysis pubis by the puboprostatic ligaments. The force of the trauma causes a displacement of pelvic bones capable of inducing urethral rupture by shearing mechanism. This force is transmitted to the prostato–membranous junction and can damage the external urinary sphincter, common penile arteries, and cavernous nerves.

Initial medical management should focus on the stabilization of the patient, as the pelvic fracture is often accompanied by vital organ injuries. A urethral injury should be suspected in the presence of hematuria or blood at the urethral meatus. Once identified, the goal of treatment should be urinary drainage to prevent extravasation and infection [3,4].

Urinary diversion can be performed through suprapubic cystostomy placement or by primary endoscopic or open realignment. The ideal management remains controversial, but some studies advise against open realignment due to the high risk of urinary incontinence and erectile dysfunction. Nevertheless, an open approach may be necessary for cases with concomitant bladder or rectal injury [5,6].

Following the acute management of the patient, it is recommended to wait 3 to 6 months to perform urethroplasty. This allows for the hematoma and inflammation to resorb. It is important that this surgery is performed by a urologist with expertise in urethral reconstruction [7]. The key to success is excising all scar tissue and performing the urethral anastomosis without tension [8,9].

In the majority of cases, the defect between the urethral ends is small and the ideal treatment is an end-to-end urethroplasty via the perineal route. In most cases, mobilization of the bulbar urethra up to the penoscrotal junction will allow performing a spatulated tension-free anastomosis. In cases with extensive defects, sequential maneuvers exist to achieve a tension-free anastomosis [10]. First is urethral mobilization (Figure 1) and separation of the corpora cavernosa, also known as crural separation (Figure 2A,B). Beginning at the level of the crus and progressing distally about 4–5 cm, this separation allows to position of the urethra between the corpora cavernosa, hence shortening the distance to the proximal urethral end. If this is not sufficient, an inferior pubectomy may be performed (Figure 3). This allows the urethra to be redirected cephalad, providing an additional 2 cm. Finally, if these maneuvers are not sufficient, supracrural rerouting is performed by passing the urethra around the lateral side of the left or right corporal body (Figure 4A,B). This shortens the distance of the anastomosis by up to 2 cm.

The aim of our study is to describe our series of urethral strictures caused by blunt perineal trauma and to analyze the impact of different preoperative variables on surgical outcomes.

## 2. Materials and Methods

A retrospective review of patients that underwent urethroplasty for urethral disruption or strictrure following blunt perineal trauma was performed. The patients were operated on by two expert surgeons in urethral surgery at our center between 1965 and 2020.

Some patients had been referred from outside institutions with a diagnosis of stenosis or urethral disruption. Many of the patients had also undergone previous failed urethral repair attempts.

Demographic data, the delay between trauma and surgery, mechanism of action of the trauma, emergency management, previous urethral interventions, surgical technique carried out in our center, complications, presence of erectile dysfunction and urinary incontinence were collected.

Prior to surgery, a physical examination along with a retrograde and voiding cystography was performed (Figure 5). Patients with no bladder catheter also underwent flowmetry. All patients underwent deferred perineal urethroplasty, and some required a combined approach. All but three were anastomotic urethroplasties. Surgical data was collected, including the use of an auxiliary maneuver as described previously to achieve a tension-free anastomosis.

After surgery, voiding urethrography with a catheter was performed in all patients prior to its removal. If urinary leakage was evident, the bladder catheter was maintained and urethrography was repeated one week later if the leak was small. Antibiotic prophylaxis was maintained from surgery until three days after catheter removal. In some patients, the suprapubic cystostomy was removed during surgery, and in others, after the removal of the bladder catheter, this was at the discretion of the surgeon.

Follow-up with flowmetry was performed at 3, 6, and 12 months during the first year, and only in cases of suspected failure was urethroscopy or retrograde and voiding urethrography performed.

Patients who did not need additional interventions during follow-up were considered successful. Data on urinary continence and erectile dysfunction were also collected. Patients that did not require the use of pads were considered continent. Sexual function was determined subjectively, according to the data provided by the patient.

For data analysis, categorical variables were expressed as frequencies and percentages. Quantitative variables were described as means and standard deviations (SD) or medians and interquartile ranges (IQR) in case of normal or non-normal distribution, respectively. The Kolmogorov–Smirnov test was used to evaluate the sample distribution. Bayesian parametric statistics and the Chi-square test for nonparametric evaluations were used for data analysis. All tests were performed setting significance at *p* < 0.05. The IBM SPSS Statistics, Version 23.0 (IBM Corp., Armonk, NY, USA), was used for the statistical analysis.

## 3. Results

A total of 116 patients were analyzed, all of whom underwent urethroplasty for traumatic urethral stricture between 1965 and 2020 at La Paz University Hospital. All patients were male, with a mean age of 33 years (±14.5).

Pelvic fracture was associated with injury to a vital organ in 82 patients (70.7%). The most frequent cause of trauma was traffic accidents (79 patients, 68%), followed by crushing injury (28 patients, 24%). Some 31 patients did not present with pelvic fracture, and 18 (58%) of these patients had suffered a straddle injury as the most frequent cause of trauma, followed by crushing and closed perineal trauma (10 patients, 32.5%). A further 43 patients were treated in the emergency setting elsewhere and were later referred to our center with the diagnosis of urethral disruption or stricture.

Fifty-two (50.5%) patients were treated initially with suprapubic cystostomy and 18 (17.5%) underwent urethral realignment, 13 (12.6%) were endoscopic realignment and 5 (4.9%) were open realignment. 8 (7.8%) patients underwent transurethral urinary catheterization in the emergency room.

Twenty-one (20.4%) patients did not require any kind of emergency management. The most common form of trauma in this group was straddle injury. Only 5 (23.8%) of these patients had a pelvic fracture.

Four patients underwent anastomotic urethroplasty in the acute setting. Two patients had pelvic fracture requiring a combined approach and the other two had a large scrotal and perineal hematoma after a straddle fall requiring evacuation and hemostasis.

The mean length of the urethral defect was 2.2 cm (±1.1), affecting the membranous urethra in 78 (67%) cases. The location of the urethral injury is summarized in Table 1.

In total, 44 (37.9%) cases had a history of previous urethral manipulation (urethroplasty, urethrotomy, or dilatation). Thirty-two (27.6%) of all patients had undergone previous urethroplasty, while 20 (17.2%) had undergone a previous urethrotomy.

All but 2 patients underwent anastomotic urethroplasty. In order to achieve a tension-free anastomosis, separation of the corpora cavernosa was required in 71 (61.5%) of the cases and partial pubectomy in 22 (18.75%). Urethral rerouting was not necessary for any patient (Table 2).

The Foley catheter was maintained for an average of 21 days (±10). Prior to bladder catheter removal, urethrography was performed in all patients to assess urinary leakage. Erectile dysfunction of any degree was reported in 40.5% of patients after the trauma, and 6 of these patients required penile prosthesis placement. New onset of erectile dysfunction was noted in 5 (4.3%) patients after anastomotic urethroplasty.

Only two patients developed urinary incontinence after surgery, one of them leaking at high bladder volumes only, and the second required placement of an artificial urinary sphincter. Two patients developed overactive bladder requiring botulinum toxin injection.

After surgery, the success rate was 91.3% (105 patients), with a median follow-up of 16 (6–47) months and a median peak urinary flow of 21 (15–30) mL/s. Of the 10 patients who relapsed, 3 underwent repeat urethroplasty, 4 underwent internal urethrotomy, and 3 underwent dilatation, ultimately achieving success in all patients.

Different factors were analyzed to see their relationship with surgical failure. No statistically significant associations were found between the history of previous urethral manipulation, the presence of concomitant pelvic fracture at the time of trauma, or the length of urethral defect with respect to failure. In patients who required placement of a suprapubic catheter in the emergency room, a tendency towards less successful surgery was observed. However, this result was not statistically significant (*p* = 0.082).

The correlation between the length of the urethral defect and the need for the placement of a suprapubic catheter in the emergency department was analyzed, but no significant association between the two was detected. Furthermore, patients with pelvic fractures did not have longer strictures nor a greater history of urethral manipulation compared to those without pelvic fractures.

## 4. Discussion

In this present study, we describe our experience in traumatic urethral strictures caused by blunt perineal trauma and analyze the impact of a number of preoperative variables on surgical outcomes. It is not completely clear what is the best acute management, nor how it may affect the complexity and success of subsequent anastomotic urethroplasty. Immediate urethral realignment or cystostomy placement are both recommended by the EAU guideline [3]. Thereafter the patient may be treated with delayed anastomotic urethroplasty 3 to 6 months later. In patients with limited prostatic displacement and without important head or pelvic trauma precluding general anesthesia, endoscopic urethral realignment can be performed during the first 1–2 weeks following the placement of the emergency cystostomy.

Primary endoscopic realignment has a lower rate of stricture formation compared to suprapubic cystostomy. If a stricture develops, they are usually shorter, making subsequent repair easier [11,15,16]. Although primary urethral realignment offers good results in approximately 30% of the patients, it can be unavailable due to limited access to specialized care and can be impractical in hemodynamically unstable patients or when the gap between the urethral ends is too long to attempt realignment. In these cases, a suprapubic cystostomy is performed. Almost 100% of these patients will develop a urethral stricture [15,17].

The performance of cystostomy involves less blood loss and less manipulation. However, the main disadvantage is that virtually all patients will develop a stricture [18]. Mouraviev et al., in their study, 96 patients reported lower stricture rates in patients managed with primary realignment (49% vs. 100%) [19].

Koraitim et al. and Fu et al. found that after endoscopic realignment, patients had less fibrosis, shorter urethral defects, and that urethroplasty was less complex compared to those who underwent cystostomy [20,21]. In the same manner, Zou et al. showed that patients managed with immediate realignment required fewer auxiliary maneuvers during urethroplasty (25.6% vs. 60.3%) [18].

In the present study, patients managed with suprapubic cystotomy had a longer mean stricture length compared to the rest of the patients (2.4 ± 1.1 cm vs. 2.1 ± 1 cm), however, this difference was not statistically significant. Anastomotic urethroplasty via the perineal route is the treatment of choice for bulbar strictures ranging from 2–3 cm. Urethral defects of up to 7–8 cm caused by disruption can also be treated perineally using the elaborated perineal posterior urethroplasty, which comprises the four steps previously described: urethral mobilization, crural separation, inferior wedge pubectomy, and supra-crural urethra rerouting. The reported success rate is around 85–90%.

Culty et al. showed in their study that patients with previous urethral surgery (urethrotomy or failed urethroplasty) had a significantly lower success rate compared to those who did not (60% vs. 90%) [22]. Similar results were obtained by Singh et al. [23], with the caveat that two or fewer endoscopic interventions did not influence the urethroplasty success rate. Other studies found no relationship with previous urethral manipulation [12]. This is the case of our study, where no relationship was found between previous urethral manipulation and surgical failure (*p* = 0.128).

Although the preferred surgical approach is the elaborated perineal posterior urethroplasty, sometimes, due to surgical complexity, fibrosis, or long stricture length, a combined perineal and abdominal approach is required. In the meta-analysis performed by Johnsen et al., the abdominoperineal approach and urethral rerouting were more frequent in cases that had been initially managed with suprapubic cystostomy [12].

In the series of Koraitim et al. [9], a combined approach was reserved for complex cases with defects larger than 2.5 cm. In our study, 93.3% of the patients requiring the combined approach had complex pelvic fractures and had a median length of 2.5 cm (1.5–5). The use of the auxiliary steps of the elaborated perineal posterior urethroplasty varies with different authors (Table 2).

In a study by Kizer et al. [11], 66% of patients did not require accessory maneuvers, and only 2.8% required separation of the corpora cavernosa. On the contrary, Flynn et al. [10] found that only 8% did not require accessory maneuvers. Thirty-four percent required separation of the corpora cavernosa, 12% required inferior pubectomy, and 38% urethral rerouting (Table 2).

Lastly, in Johnsen et al.’s study, 64% of cases did not require auxiliary maneuvers and direct urethral anastomosis was performed. Of the remaining patients, 36% underwent crural separation, 11% underwent inferior wedge pubectomy, 2% underwent complete pubectomy, and 2% underwent supracrural rerouting [12]. In our study, it was necessary in 61.5% of the cases to separate the corpora cavernosa in order to perform a tension-free anastomosis, 18.75% of patients needed inferior wedge pubectomy, and none required urethral rerouting. However, we acknowledge that urethral rerouting is a very useful maneuver in patients with very long urethral defects, and we have used it several times in other scenarios (Table 2).

Despite the complexity of these patients, high success rates have been described. Our success rate reached 91.3% with a mean follow-up of 45 ± 79 months. In the study by Johnsen et al. [12], similar success rates were observed, averaging 90%. Kulkarni et al., in their study of 92 cases, found lower success rates compared to Barbagli’s series of 120 cases (78.6% vs. 86.9%), which may be due to the greater severity of the initial trauma in India [24].

In patients who recur, short stenotic rings are usually found in the area of the anastomosis. In these cases, endoscopic urethrotomy is a possible management option. Netto et al. reported a 72% success rate with internal urethrotomy for recurrent strictures after anastomotic urethroplasty [25]. Morey et al. [26] found a slightly higher success rate averaging 88%. In our study, we found a 100% success rate after the use of endoscopic urethrotomy.

This study has several limitations. First, it is a retrospective review with a relatively low number of cases. Post-traumatic urethral stricture is a very rare entity and is seen more frequently in the developing world. Secondly, two cohorts of patients were used, the data of the first one was collected between 1965 and 1999 [27], and the second cohort between 2000 and 2020. This explains why the follow-up in both cohorts is limited. Thirdly, emergency management was not standardized, as many patients were treated in the emergency setting elsewhere and were later referred to our center with a diagnosis of urethral stricture. On the other hand, as it is a complex surgery, it must be performed by experienced urologists. In our study, patients were operated on by two expert reconstructive surgeons; this could explain why our high success rate might not be applicable to other centers.

## 5. Conclusions

Elaborated perineal posterior urethroplasty for traumatic urethral defects and strictures is a complex surgery, but it can achieve a high success rate when performed by experienced reconstructive urologists.

Corporal body separation is required in a high percentage of cases. Urethral rerouting was not required in any of our patients, although we acknowledge that it is a very useful maneuver in patients with very long urethral defects.

## Figures and Tables

**Figure 1 jcm-12-00054-f001:**
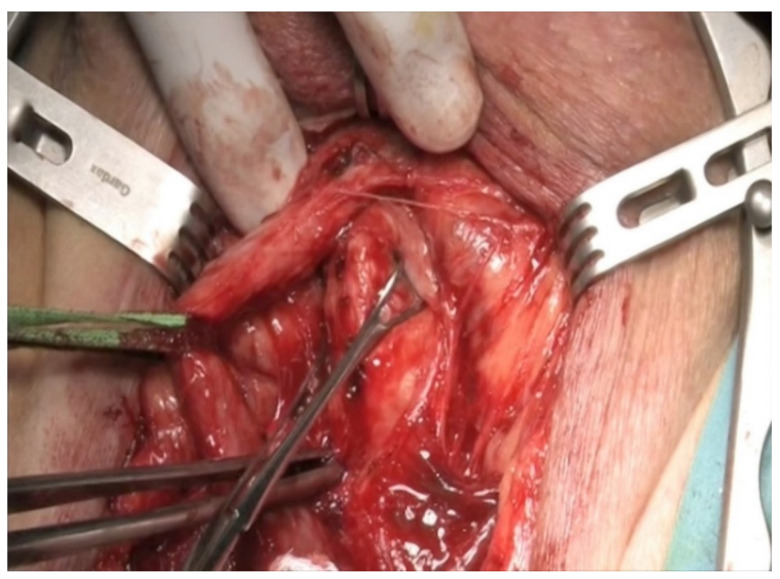
Urethral mobilization.

**Figure 2 jcm-12-00054-f002:**
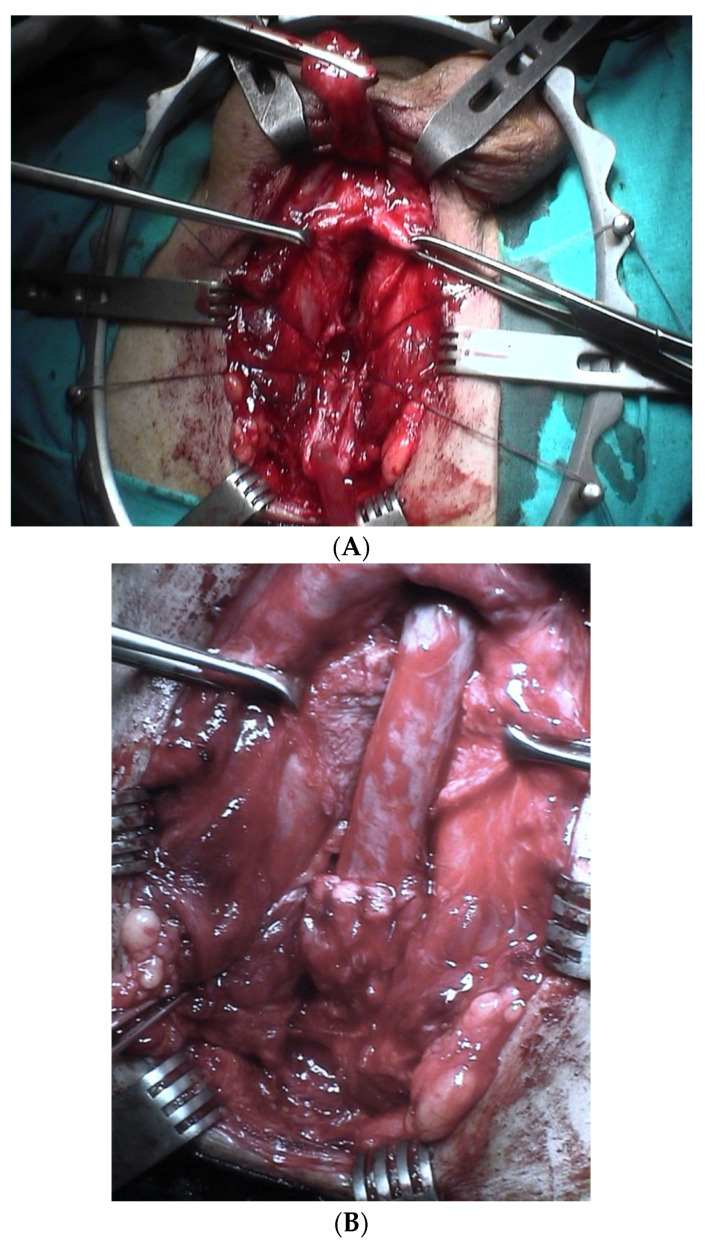
(**A**,**B**) Corporal body separation.

**Figure 3 jcm-12-00054-f003:**
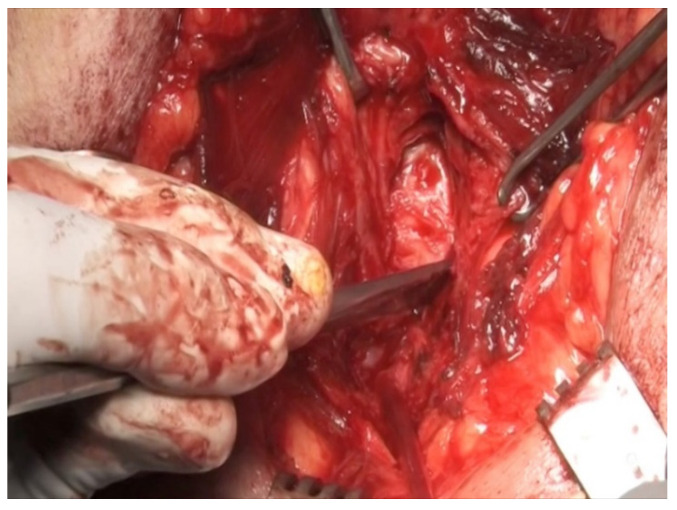
Inferior wedge pubectomy.

**Figure 4 jcm-12-00054-f004:**
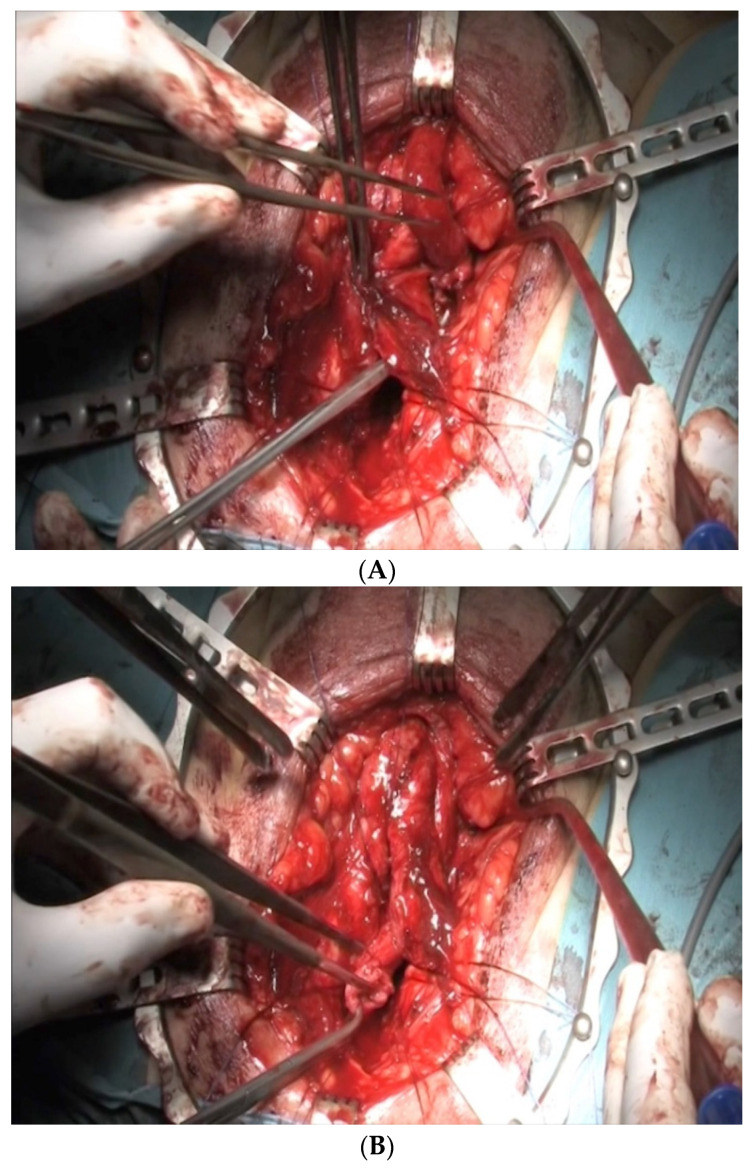
(**A**,**B**) Urethral re-rounting.

**Figure 5 jcm-12-00054-f005:**
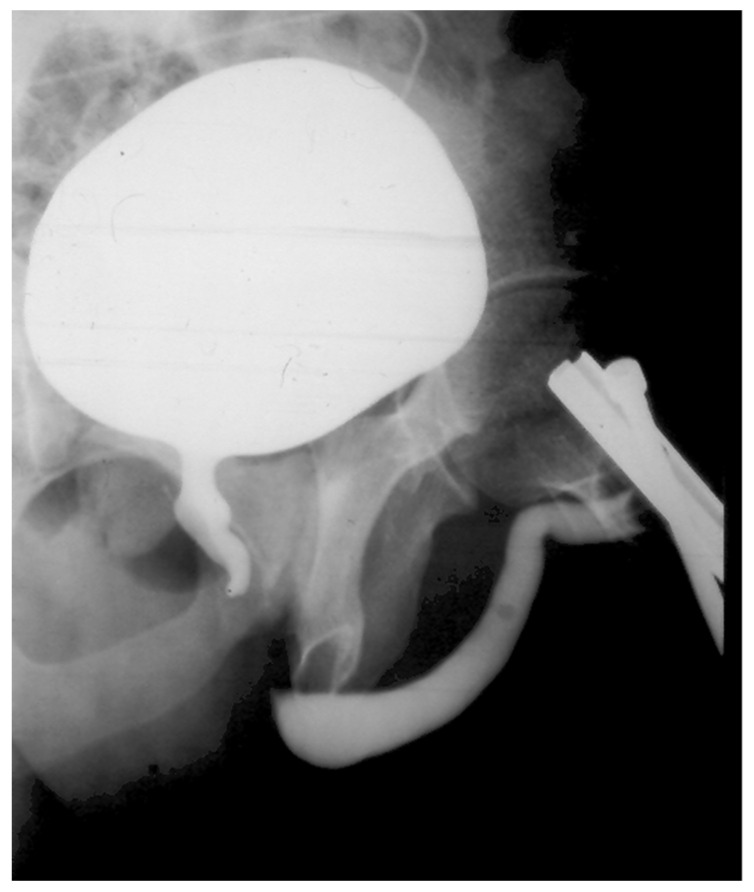
Cystourethrography in a patient with urethral disruption.

**Table 1 jcm-12-00054-t001:** Location of the urethral injury in 116 cases.

Membranous Urethra	Membranous Urethra and Proximal Bulbar Urethra	Only Proximal Bulbar Urethra	Mid Bulbar Urethra	Other Sites
21 (18.1%)	57 (49.1%)	30 (25.9%)	4 (3.4%)	4 (3.4%)

**Table 2 jcm-12-00054-t002:** Percentage of cases using the different steps of the elaborated perineal posterior urethroplasty by authors.

	*n*	Urethral Mobilization	Corporal Body Separation	Inferior Wedge Pubectomy	Urethral Re-Rounting	Success Rate
Kizer et al. [11]	142	67%	17%	10%	3%	91.5%
Flynn et al. [10]	122	8%	34%	12%	38%	92%
Johnsen et al. [12]	122	64%	36%	11%	2%	91%
Webster et al. [13]	74	11%	44%	30%	15%	96%
Fu et al. [14]	301	34.2%	29.6%	31.6%	4.7%	87.4%
Our series	116	100%	61.5%	18.75%	0%	91.3%

## Data Availability

The data are available upon request from the corresponding author.

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
