# Peer review of "Repair of Traumatic Urethral Strictures: La Paz University Hospital Experience"

_jcm, 2022, doi:10.3390/jcm12010054_

Round 1

Reviewer 1 Report

The authors report on their experience with traumatic urethral strictures. This is an interesting paper with a few shortcomings.

1) the cohort is very heterogenous for example in respect to trauma etiology or whether realignement was done. A subanalysis would improve this paper but may not be possible.

2) there is a lot of disagreement whether primary relignement is useful. The authors focus on studies that show its advantage but need to include those that are critical of it. Realignement will not prevent a stricture but it makes the subsequent repair easier.

3) The Discussion needs to be streamlined. There is no need to quote detailed outcomes taken from other papers, a sentence or two with conclusions is sufficient.

Author Response

Greetings
Thank you very much for your review. 
First of all, as this is a rare pathology, unfortunately we have a very heterogeneous cohort and many patients were referred from other centers, making it impossible to perform a subanalysis of urgency management.
Secondly, we have mentioned the possibility of performing urethral realignment and suprapubic cystostomy as emergency management. We describe both managements emphasizing that there is no ideal management, the success rate of urethral realignment is only 30% and objective data are given from a meta-analysis performed by Barret.
Thirdly, we find it very interesting to show the results of other studies with a number of patients similar or superior to ours, since there are not many published works on this subject. We believe that this is a good summary of the data provided by other authors and that the reader may find it of interest.

Regards.

Reviewer 2 Report

The authors have presented a retrosepctive cohort series of 116 patients spanning over 55 years with repair of urethral injuries at one of the advanced tertiary urology center by two urologist with special interest in urethral reconstruction with suprb results. Overall the presentation is good and mahority patients being complex and challenging is useful to the readers. However, the presentation can be improved by minor changes. In the Introduction section first two and last paragraphs can be kept, paragraphs 3-7 can be moved to the beginning of the Discussion section and remaining paragraphs 8-9 can be moved to Methods section after figure 1 and before the paragraph begining with After surgery... so that it will fit well with the diagnostic evaluation and post operative managemnt and the figure sequence will get maintained in order. The references section can be improved by removing very old references and replacing them with the recent ones.

Author Response

We appreciate your review, and will therefore proceed to make the changes you advise us to make. Thank you very much.

Round 2

Reviewer 1 Report

This is a well-written paper. The main concern is novelty as urethroplasties for traumatic strictures has been described albeit not from that geographic location. The fact that case as far back as 1965 are included is standing out. This paper would benefit from a report on how the surgical technique changed (and likely evolved) over the past 6 decades.

Author Response

Thank you very much for the review. 
The surgical technique in our center has not changed over the years. In all patients a termino-terminal urethroplasty was performed using the different maneuvers to perform the anastomosis without tension. 
We have not seen greater complexity in the cases or the need for more auxiliary maneuvers in the patients operated on before 2000. 
On the other hand, it has been seen that the number of patients with traumatic stenosis has decreased over the years due to development and that nowadays traumatisms are less frequent. 

Thank you